# Lipoprotein (a) as a Cardiovascular Risk Factor in Controversial Clinical Scenarios: A Narrative Review

**DOI:** 10.3390/ijms252011029

**Published:** 2024-10-14

**Authors:** Hesham M. Abdalla, Ahmed K. Mahmoud, Ahmed E. Khedr, Juan M. Farina, Isabel G. Scalia, Mohammed Tiseer Abbas, Kamal A. Awad, Nima Baba Ali, Nadera N. Bismee, Sogol Attaripour Esfahani, Niloofar Javadi, Milagros Pereyra, Said Alsidawi, Steven J. Lester, Chadi Ayoub, Reza Arsanjani

**Affiliations:** 1Department of Internal Medicine, Mayo Clinic, Phoenix, AZ 85054, USA; abdalla.hesham@mayo.edu; 2Department of Cardiovascular Medicine, Mayo Clinic, Phoenix, AZ 85054, USAfarina.juanmaria@mayo.edu (J.M.F.); scalia.isabel@mayo.edu (I.G.S.); abbas.mohammedtiseer@mayo.edu (M.T.A.); awad.kamal@mayo.edu (K.A.A.); bismee.naderanaquib@mayo.edu (N.N.B.); attaripouresfahani.sogol@mayo.edu (S.A.E.); javadi.niloofar@mayo.edu (N.J.); pereyra.milagros@mayo.edu (M.P.); alsidawi.said@mayo.edu (S.A.); lester.steven@mayo.edu (S.J.L.); ayoub.chadi@mayo.edu (C.A.); 3Department of Cardiothoracic Surgery, Mayo Clinic, Phoenix, AZ 85054, USA; khedr.ahmed@mayo.edu

**Keywords:** cardiac allograft vasculopathy, lipoprotein(a), atrial fibrillation, in-stent restenosis, bioprosethetic aortic valve degeneration

## Abstract

Lipoprotein (a) is a complex lipid molecule that has sparked immense interest in recent years, after studies demonstrated its significant association with several cardiovascular conditions. Lp(a) promotes cardiovascular disease through its combined proatherogenic, pro-inflammatory, and prothrombotic effects. While the measurement of Lp(a) has become widely available, effective methods to reduce its concentration are currently limited. However, emerging data from ongoing clinical trials involving antisense oligonucleotides have indicated promising outcomes in effectively reducing Lp(a) concentrations. This may serve as a potential therapeutic target in the management and prevention of myocardial infarction, calcific aortic stenosis, and cerebrovascular accidents. In contrast, the role of Lp(a) in atrial fibrillation, in-stent restenosis, cardiac allograft vasculopathy, and bioprosthetic aortic valve degeneration remains unclear. This review article aims to thoroughly review the existing literature and provide an updated overview of the evidence surrounding the association of Lp(a) and these cardiovascular diseases. We seek to highlight controversies in the existing literature and offer directions for future investigations to better understand Lp(a)’s precise role in these conditions, while providing a summary of its unique molecular characteristics.

## 1. Introduction

Despite notable advancements in the treatment and prevention of atherosclerotic cardiovascular disease, it remains the leading cause of morbidity and mortality, globally [1]. While this suggests a high prevalence of traditional cardiovascular risk factors, residual cardiovascular risk (defined as the risk of cardiac events that persists despite reaching targets for traditional risk factors) remains high. Therefore, there has been an increasing focus on identifying additional the contributing factors that may be involved. Lipoprotein (a) [Lp(a)] is a liver-derived lipoprotein, which was first discovered by geneticist Kare Berg, in 1963 [2]. Several decades after its discovery, epidemiological and genetic studies demonstrated a causal association between elevated concentrations of Lp(a) and ischemic heart disease [3]. More recently, studies have established associations between elevated serum Lp(a) concentrations and the risk of myocardial infarction, calcific aortic stenosis (AS), and cerebrovascular accidents [4]. The current American College of Cardiology/American Heart Association guidelines suggest that levels ≥ 50 mg/dL (or ≥125 nmol/L) are considered abnormal and classified as a risk-enhancing factor for cardiovascular disease [5] (Table 1).

A growing number of studies have attempted to compare the cardiovascular risks posed by low density lipoprotein (LDL) and Lp(a). Björnson et al. concluded that, while LDL particles were more abundant in the serum, Lp(a) was approximately six times more atherogenic on a per particle basis [6]. In a retrospective study of pediatric patients, those with familial hypercholesterolemia and a family history of early ASCVD were three times more likely to have higher Lp(a) levels (OR: 3.77, 95% CI: 1.16–12.25, *p* = .027), whereas no significant correlation was observed with LDL cholesterol levels (OR: 0.45, 95% CI: 0.11–1.80, *p* = .26) [7]. 

In relation to myocardial infarction, a study of approximately 100,000 participants from the Copenhagen General Population Study identified Lp(a) and LDL as equal causal contributors [8]. Additionally, the combined elevation of both LDL-C and Lp(a) was associated with a higher risk of first acute myocardial infarction compared to the sum of their individual risks, suggesting a potential synergistic mechanism [9]. These studies suggest that Lp(a) and LDL contribute equally as causal factors for myocardial infarction [10]. Numerous studies have demonstrated a positive association between Lp(a) levels and cerebrovascular disease [11]. Interestingly, when comparing Lp(a), cholesterol, LDL, high density lipoprotein (HDL), and triglycerides, only Lp(a) and cholesterol were identified as independent risk factors of carotid atherosclerosis [12]. However, research remains limited and further studies are required to compare Lp(a) with other lipid particles regarding stroke risk. 

Strategies to lower Lp(a) concentrations are limited as traditional lipid modifying therapies, such as statins and ezetimibe, demonstrated a minimal impact on Lp(a) concentrations [13]. However, recent advancements, including the evaluation of antisense therapy targeting the mRNA of apolipoprotein(a) [apo(a)], have sparked immense interest and an evolving body of literature on Lp(a) as a risk factor for cardiovascular diseases. As these therapies show promise in significantly lowering Lp(a) concentrations, evaluating the associations between Lp(a) and various cardiovascular conditions has become increasingly important. While its association with certain cardiovascular conditions is well-established, there are several key conditions for which evidence of their association remains weak or controversial. 

**Table 1 ijms-25-11029-t001:** Scientific Society Guidelines on Lipoprotein (a) Testing.

Scientific Society	Screening Indications/Recommendations	Cut-Off Values
ACC/AHA—Guideline 2019 [5]	- A relative indication for its measurement is family history of premature ASCVD. - Classed as a “risk-enhancing factor” in patients 40 to 75 years old, without diabetes mellitus but with 10-year ASCVD risk–7.5 to 19.9% would favor initiation of statin therapy.	>50 mg/dL (125 nmol/L)
ESC/EAS—Guideline 2019 [14]	- Measure in all patients at least once in their adult lifetime to identify those who may have a very high lifetime risk of atherosclerotic cardiovascular disease (ASCVD), similar to those with heterozygous familial hypercholesterolemia (FH).	>180 mg/dL (430 nmol/L) [very high risk] - Normal value not specified
National Lipid Association (NLA)—Scientific Statement 2021 [15]	Reasonable to refine ASCVD risk assessment in adults with: - first-degree relatives with premature ASCVD (<55 years of age in men and <65 years of age in women); - a personal history of premature ASCVD; and - primary severe hypercholesterolemia or suspected FH.	>50 mg/dL (125 nmol/L)
Canadian Cardiovascular Society (CCS)—Guideline 2021 [16]	Recommend measuring Lp(a) level once in a person’s lifetime as a part of the initial lipid screening.	>50 mg/dL (125 nmol/L)
European Atherosclerosis Society (EAS)—Consensus Statement 2022 [17]	Lp(a) should be measured at least once in adults to identify those with high cardiovascular risk.	>50 mg/dL (125 nmol/L)

This review aims to thoroughly review the existing literature and provide an updated overview of the evidence surrounding the association of Lp(a) and atrial fibrillation, in-stent restenosis, cardiac allograft vasculopathy, and bioprosthetic aortic valve degeneration. We seek to highlight controversies in the existing literature and offer directions for future investigations to better understand Lp(a)’s precise role in these conditions (Figure 1).

## 2. Results

A total of 21 articles were included in this review, addressing the association between Lp(a) and atrial fibrillation, in-stent restenosis, cardiac allograft vasculopathy, and bioprosthetic aortic valve degeneration. Details of the studies included are provided in Appendix A. 

### 2.1. Lp(a) and Atrial Fibrillation

Several research studies have identified an intriguing inverse association between elevated Lp(a) levels and Atrial Fibrillation (AF) risk, challenging the conventional understanding of AF risk factors. In a retrospective cohort analysis involving 13,522 patients within a Chinese population, Tao et al. demonstrated that lower median Lp(a) levels were associated with a significantly higher prevalence of AF (15.95 mg/dL in AF patients vs. 16.90 mg/dL in controls; *p* < 0.001) [18]. Additionally, the authors suggested that an Lp(a) level lower than 32.42 mg/dL could serve as a potential risk factor for developing AF. Importantly, a further stratified analysis revealed that a significant association was observed only in women and was not evident in patients with coronary artery disease (CAD), ischemic stroke, and type 2 diabetes mellitus. 

Garg et al. conducted a large multiethnic cohort study involving 6593 participants, demonstrating that Lp(a) levels ≥ 30 mg/dL were inversely associated with AF risk over a median follow-up period of 29 years (HR 0.84; 95% CI 0.71–0.99; *p* = 0.035) [19]. A subgroup analysis additionally revealed that this association was specific to the Chinese population, and no association was found with White, Black or Hispanic populations. Similarly, Xie et al. found that elevated Lp(a) were inversely associated with AF risk in a Chinese population [OR 0.94; 95% CI, 0.901–0.987; *p* = 0.012)] [20]. 

Conversely, numerous studies have reported a significant association between elevated Lp(a) and increased AF risk. Analyzing data from 20,432 patients in the UK Biobank, Shemirani et al. found that higher Lp(a) levels were significantly associated with an increased risk of AF (HR: 1.03; 95% CI: 1.02–1.04; *p* < 0.01), implicating Lp(a) as a potential causal mediator [21]. A meta-analysis of mendelian randomization (MR) studies also demonstrated a positive association (OR 1.024; 95% CI: 1.007–1.042; I^2^ = 87.72%; *p* < 0.001), highlighting a greater risk in European populations [17]. An MR analysis similarly concluded that genetically elevated levels of Lp(a) increased the risk of AF (OR [95% CI] = 1.001 [1.000–1.002]; *p* = 0.016) [22]. 

In contrast, Aronis et al. investigated the ARIC (atherosclerosis risk in communities) cohort, which included only white and black participants, and found no association between Lp(a) levels and the risk of AF [23]. Mora et al. documented similar results in a cohort of 23, 738 relatively healthy women, observing no association between Lp(a) levels and AF risk over a median follow-up of 16.4 years [24]. 

### 2.2. Lp(a) and In-Stent Restenosis

A number of observational studies have investigated the relationship between Lp(a) levels and In-stent restenosis (ISR), yielding some contradictory but generally supportive findings of a positive association. One of the first studies to assess this relationship included 2223 patients from 1993 to 1997, who had elevated Lp(a) and who had undergone stent placement [25]. The study found no significant influence of elevated Lp(a) on adverse one-year clinical and angiographic outcomes after stent placement. Restenosis occurred in 173 out of 523 lesions (33.2%) in the elevated Lp(a) group and 636 out of 1943 lesions (32.7%) in the low Lp(a) group (OR; 1.02, 95% CI; 0.83–1.25, *p* = 0.88). Ribichini et al. specifically investigated this relationship in 325 patients who had Lp(a) measurements before percutaneous coronary intervention (PCI), with angiographic follow-up performed at 6 months [26]. Baseline Lp(a) was not a predictor for restenosis after elective coronary stenting. Similarly, Khosravi et al. did not demonstrate an association, conducting a study that included 170 patients with follow-up coronary angiography 6 months after baseline angioplasty (OR; 0.54, 95% CI; 0.26–1.10; *p* = 0.09) [27]. 

On the other hand, a meta-analysis of nine cohort studies, including 1,834 patients (600 ISR and 1234 non-ISR patients), found significantly elevated baseline Lp(a) levels in ISR patients (SMD = 0.42; 95% CI: 0.14–0.71; *p* = 0.003). This finding was most notable in the Asian population [28]. Furthermore, a study conducted on 595 Asian patients who underwent elective PCI with drug-eluting stents (DES) demonstrated that ISR incidence was higher in the high Lp(a) group (≥50 mg/dL) compared to the lower Lp(a) group (19.8% vs. 7.9%, *p* = 0.001). In a multivariable model, high Lp(a) levels were significantly associated with ISR (aHR: 2.88, 95% CI: 1.36–6.07, *p* = 0. 005) [29]. 

Kamitani et al. conducted a prospective study, which included 109 patients who had successfully undergone elective coronary stent implantation, with follow-up angiography at 24 ± 6 weeks. Lp(a) levels were higher in the restenosis group (30.5 ± 23.9 vs. 16.9 ± 11.1 mg/dL, *p* < 0.01). In a multivariable analysis, Lp(a) remained significant as an independent predictor of restenosis (aOR: 1.37; 95% CI: 1.05–1.79; *p* = 0.02) [30]. 

A recent study that included 119 patients with DES ISR who underwent PCI guided by optical coherence tomography, showed a significantly higher incidence of in-stent neoatherosclerosis in the high Lp(a) group (94.0% vs. 52.0%, *p* < 0.001). Moreover, the incidence rate of thin-cap fibroatheroma in ISR lesions was significantly higher in the high Lp(a) group (42% vs. 5.3%, *p* < 0.001), indicating higher plaque vulnerability [31]. The long-term effects of Lp(a) on ISR were demonstrated by a recent retrospective study involving 1,209 patients. High Lp(a) levels were significantly associated with ISR (aHR: 1.67, 95% CI: 1.18–2.37, *p* = 0.004). A landmark analysis showed that the association gained significance after the first year post-PCI and became more pronounced after 2–3 years, indicating a potential long-term effect [32]. 

### 2.3. Lp(a) and Cardiac Allograft Vasculopathy

In a recent observational study including 150 HTx patients under follow up in a tertiary center, a univariate and multivariate analysis was performed to evaluate the association between Cardiac allograft vasculopathy (CAV) and Lp(a) levels, donor’s age, LDL cholesterol levels, triglyceride levels, and cytomegalovirus (CMV) infection. The only factor found to be associated with CAV Grade 2 and 3 (based on the International Society for Heart and Lung Transplantation’s cardiac allograft vasculopathy grading scheme) was elevated Lp(a) levels defined as Lp(a) >30 mg/dl (*p* < 0.001 and an OR of 8.57 [95% CI 2.82–19.55]). To be included in the study, all patients had to have had at least one coronary angiography after transplantation, which was performed one year after transplantation or in case of clinical suspicion of CAV. However, since only one coronary angiogram was performed in the follow-up period, CAV incidence may have been underestimated [33]. Similarly, Barbir et al. reported in their cross-sectional study of 130 heart transplant patients that elevated Lp(a) concentrations showed an independent significant association with the development of accelerated CAV after transplantation. The median Lp(a) level of the 33 patients with graft CAV was 71 mg/dl (IQR 3,196) vs. 22 (IQR 1,170) in patients without CAD (*p* < 0.001) [34].

Yet, the role of Lp(a) in CAV remains controversial. In a retrospective cohort study of 74 transplant patients, no significant difference was found between Lp(a) levels in patients with or without CAV. Mean Lp(a) levels (20 mg/dl ± 19 [CAV (+)] vs. 30 mg/dl ± 30 [CAV (–)]; *p* = NS) did not differ between the patients with and those without CAV. Other baseline characteristics were also assessed, including LDL cholesterol levels up to one year after transplant, triglyceride levels up to one year after transplant, and donor characteristics (age, race, and sex). Out of the 74 patients followed, 25 demonstrated angiographic evidence of CAV, with the only independent predictor of CAV being advanced donor age. However, it is important to note that this study did not standardize the use of lipid-lowering medications, which may have influenced the development of CAV [35]. One limitation in all the studies previously mentioned was the lack of baseline angiographies (<3 months after transplant) to rule out the donors’ pre-existing CAD. 

### 2.4. Lp(a) and Bioprosthetic Aortic Valve Degeneration

Current studies evaluating the association between Lp(a) and Bioprosthetic heart valves (BHV) are fairly limited. Farina et al. conducted a retrospective analysis of 210 patients who underwent BHV placement across three academic centers. Over a median follow up of 4.4 years, echocardiographic data identified 33 patients that developed SVD and exhibited significantly higher Lp(a) levels compared to those without SVD [50.0 (IQR 72.0) vs. 15.6 (IQR 48.6) mg/dL, *p* = 0.002] [36]. A multivariable analysis confirmed that elevated Lp(a) levels (≥30 mg/dL) were independently associated with an increased risk of SVD (HR 4.44 95%; CI 1.89 to 10.42; *p* = 0.001). The study’s retrospective nature, limited number of SVD cases and reliance on echocardiographic data could have limited the accuracy and generalizability of its results. Nonetheless, it stands as the largest investigation to date, which directly explores this correlation. 

In contrast, Botezatu et al. investigated this association by performing a post hoc analysis of 97 patients with BHV placement, over a median follow up of two years [37]. This study utilized a multimodality imaging approach, including echocardiography, contrast-enhanced computed tomography (CT) and 18F-NaF positron emission tomography (PET) scanning to assess for early signs of SVD. No significant association was found between Lp(a) and SVD, in either imaging or clinical outcome data. Despite this, the study’s single-center focus, short follow-up period, and relatively small cohort constrained its generalizability.

## 3. Discussion

### 3.1. Lipoprotein (a) Structure and Function

Lp(a) is a complex lipoprotein similar to low density lipoprotein (LDL) but includes several key structural differences that confer its unique biochemical properties [38]. At its core, Lp(a) consists of cholesteryl esters and triglycerides, encased by a monolayer of phospholipids and unesterified cholesterol. This lipid core is encircled by apolipoprotein B100 (apoB), a hydrophilic molecule also found in LDL that ensures structural stability [39]. The presence of the lipophilic, highly glycosylated apo(a) is the hallmark of Lp(a), which differentiates it from other lipoproteins. This large glycoprotein is covalently bound to apoB via one disulfide bridge and accounts for the majority of Lp(a)’s total mass [40].

Complementary deoxyribonucleic acid (DNA) sequencing has demonstrated that apo(a) bears a strong resemblance to human plasminogen [41]. This is particularly evident in its Kringle domains, composed of triple-looped polypeptides stabilized by three internal disulfide bridges, as seen in other coagulation factors such as prothrombin and streptokinase [42]. Unlike plasminogen, which contains ten Kringle domains, apo(a) features only two types: Type 4 (K4) and Type 5 (K5). Additionally, apo(a) contains 10 subtypes of K4 (KIV_1_ to KIV_10_), with significant variations in K4 Type 2, contributing to the size heterogeneity of Lp(a) [43]. Kringle domains have unique binding sites and play essential roles in mediating pathogenic biochemical reactions that precipitate local inflammation and vascular smooth muscle proliferation [44].

The apo(a) gene (*LPA*) is located on chromosome 6q26–27 and predominantly determines Lp(a) levels and atherogenicity, largely through variation in Kringle KIV_2_ repeats [45]. Additionally, the single-nucleotide polymorphisms rs10455872 and rs3798220 in Europeans and rs9457951 in African Americans have also been linked to variations in Lp(a) levels [46,47]. 

The true physiological role of Lp(a) remains elusive, particularly in light of the considerable variation in Lp(a) levels among individuals. A study investigating a northern European population with a loss-of-function variant in the *LPA* gene observed no clinical signs or defects, further questioning its functional importance [48]. Proposed roles include modulating coagulation and fibrinolysis by inhibiting streptokinase and urokinase, thereby preventing the conversion of plasminogen to plasmin [49]. The elevated levels of Lp(a) observed in tissue during wound healing suggest that Lp(a) has a potential role in healing by serving as a cholesterol source [50]. Studies have also demonstrated elevated Lp(a) levels during acute illnesses such as myocardial infarction and inflammatory bowel disease, as well as in patients undergoing Il-6 treatment, suggesting its potential role as an acute phase reactant [51]. 

Lp(a) has been previously identified as having a prothrombotic effect. However, the physiological interaction between Lp(a) and platelets remains unclear, and its exact influence on platelet function is not well understood. Several studies have explored the impact of antiplatelet therapy on long-term cardiovascular outcomes in patients with elevated Lp(a) levels. One recent study involved 3,025 patients with coronary artery disease (CAD) and elevated Lp(a) levels, of which 913 received dual antiplatelet therapy (DAPT) for ≤1 year, while 2,112 were treated with DAPT for >1 year. The group treated for >1 year showed a significantly lower risk of major adverse cardiovascular events (MACE) compared to those treated for ≤ 1 year (1.6% vs. 3.8%; HR: 0.383, 95% CI: 0.238–0.616; *p* < 0.05) [52]. 

Additionally, another observational study focused on aspirin use in primary prevention and included 2183 patients. The findings demonstrated that aspirin use was significantly associated with a reduced risk of CAD events in patients with elevated Lp(a) (HR: 0.54, 95% CI: 0.32–0.94; *p* = 0.03) [53].

While many of the atherogenic properties of Lp(a) are mediated by apo(a), the apoB component not only provides structural integrity but also contributes to the promotion of atherosclerosis. ApoB containing lipoproteins bind to proteoglycan-binding sites through ionic interactions, which is considered to be the key mechanism driving their retention and aggregation to the arterial wall [54]. Additionally, the recognition of the ApoB100 component in oxidized LDL by CD4+ T cells aids in the progression of atherosclerosis through cytokine-mediated inflammation [55]. 

### 3.2. Lp(a) lowering therapies

In recent years, therapies aimed specifically at reducing Lp(a) levels have emerged, utilizing antisense oligonucleotides (ASOs) and small-interfering RNA (siRNA). ASOs are typically composed of DNA fragments, which are injected subcutaneously and subsequently migrate to the liver’s extracellular space. Upon binding to LPA mRNA, ASOs form ASO–mRNA complexes that are cleaved by ribonuclease H1, resulting in a decrease in Lp(a) concentrations by reducing apolipoprotein (a) production [56]. The Phase 3 Lp(a) HORIZON trial is currently evaluating the ASO Pelacarsen, with major cardiovascular events set as the primary endpoint. Phase 2 trials have yielded promising outcomes, demonstrating Lp(a) reductions of 35% with monthly dosing and 80% with weekly dosing. No hepatic adverse effects were observed, with mild injection-site reactions being the most frequently reported adverse effect [57].

siRNAs are double stranded RNA molecules, which are also directed to liver hepatocytes, where they cleave target *LPA* mRNA by utilizing the RNA-induced silencing complex (RISC). Through the inhibition of apolipoprotein (a) synthesis, this mechanism reduces Lp(a) concentrations. While early iterations of siRNA presented pharmacokinetic challenges regarding stability and delivery, recent advancements have led to the development of Olpasiran [58]. Phase 2 trials have demonstrated substantial efficacy, resulting in a −101.1% placebo-adjusted reduction in Lp(a) levels at a dose of 225 mg every 24 weeks [59]. Similarly, the most common adverse effects were injection-site reactions, which were generally transient and localized to the site of administration. The Phase 3 Ocean(a) trial is currently underway, assessing the impact of Olapsiran on major cardiovascular events. 

The findings of both the Lp(a) HORIZON and OCEAN(a) trials will offer critical insights into whether Lp(a)-lowering therapies will result in improved cardiovascular outcomes. While these studies are focused mainly on secondary prevention and high-risk populations, the development of larger studies focusing on primary prevention is both necessary and anticipated. These trials promise encouraging prospects for physicians and patients alike, introducing effective approaches to managing Lp(a)-mediated risks. Statins are likely to remain as the cornerstone of treatment for ASCVD risk reduction, supported by decades of robust evidence. However, these newer therapies could transform ASCVD management, by complimenting PCSK9 inhibitors and antithrombotic therapies in individualized and pathophysiology-based risk reduction.

### 3.3. Lp(a) and Atrial Fibrillation

AF is the most common cardiac arrhythmia observed in clinical practice, characterized by irregular atrial activity and the absence of distinct P waves on ECG. The growing prevalence of AF, fueled by increasing life expectancy and rising medical comorbidities, imposes a significant burden on public healthcare systems worldwide [60]. The early detection and prevention of AF is crucial in preventing atrial remodeling and improving long-term prognoses [61]. While numerous cardiovascular risk factors, such as hypertension, obesity, and diabetes mellitus, have been linked with an increased risk of AF, the association with Lp(a) remains unclear. 

While the relationship between Lp(a) and AF is not yet well defined, several mechanisms have been proposed. The pro-inflammatory nature of Lp(a), largely due to the transport of oxidized phospholipids, could disrupt atrial conductance and facilitate atrial remodeling and fibrosis, contributing to the progression of AF [62]. Additionally, studies have demonstrated that Lp(a) is associated with increased endothelial permeability, particularly in AS, where its infiltration of valvular tissue leads to accelerated valve calcification and cell death [63]. Although the mechanisms behind an inverse relationship between Lp(a) and AF risk remain unclear, similar associations have also been demonstrated with LDL-cholesterol and total cholesterol levels, suggesting a shared mechanism [64,65]. 

The current literature on the association between Lp(a) and AF presents conflicting and heterogenous results, hindering the ability to make definitive conclusions. Significant differences were most evident in studies evaluating specific populations, particularly between European and Chinese groups, which may partly be explained by the ethnic variation in baseline Lp(a) levels [66]. These contradictory findings also raise a possibility that additional genetic or environmental findings are implicated in the pathophysiology of AF. Variations in sample size, study design, and the presence of confounding factors likely play a role in the discrepancies in the data. Several studies in this review showed considerable differences in accounting for patients with CAD, with some studies adjusting for CAD [20,37], while others excluded CAD patients [21,39]. This likely accounts for the conflicting results encountered in this review. In addition, the lack of an association in the observational studies included may be explained by additional confounding factors, which influenced both Lp(a) levels and AF risk without being measured. 

Due to the conflicting results in the literature, a clear association between Lp(a) and AF risk cannot be established, nor can its use for assessing AF risk be recommended in clinical practice. Well-designed, large-scale studies incorporating multiple ethnic groups and the careful consideration additional confounding factors are greatly needed. Prospective clinical trials evaluating novel therapies aimed at lowering Lp(a) should include AF as an endpoint, to help determine whether lowering Lp(a) reduces the risk of AF. Clarifying the precise mechanisms between Lp(a) and AF could facilitate the development of targeted treatment and prevention strategies. Ultimately, addressing these gaps in the literature could prove to be highly beneficial in the treatment and prevention of this prevalent and burdensome arrythmia.

### 3.4. Lp(a) and In-Stent Restenosis

ISR refers to the re-narrowing of a previously stented coronary artery segment, often due to neointimal hyperplasia (Figure 2). The underlying processes include endothelial injury, vascular inflammation, smooth muscle cell proliferation, and extracellular matrix deposition [67]. The risk factors associated with the short-term development of ISR are often related to the stent itself, procedural factors (e.g., stent under-expansion) and compliance with antiplatelet therapy [67]. In the long term, the atherosclerotic process plays a crucial role in ISR, particularly through the development of neoatherosclerosis [68]. As previously mentioned, Lp(a) is a proatherosclerotic, prothrombotic, and proinflammatory molecule. These properties can trigger the atherosclerosis process, which intersects with the pathophysiology of ISR.

Patients with elevated Lp(a) levels may be at an increased risk of ISR, as suggested by the studies. However, interpreting these findings requires caution due to the continuous evolution of stent technology, ranging from bare-metal stents (BMS) to various generations of DES. The long-term impact of Lp(a) on cardiovascular risk, particularly as a trigger in the atherosclerosis process, has become a focus of ongoing research. Previous studies that showed no significant association between Lp(a) and ISR were often limited by short follow-up periods, potentially overlooking long-term effects. The Lp(a)-related risk of ISR implies that Lp(a) screening could be useful in assessing restenosis risk following stent implantation. A case report by Akiyama et al. described a 57-year-old woman with high Lp(a) levels who experienced six recurrent ISRs. After treatment with a proprotein convertase subtilisin/kexin type 9 inhibitor (PCSK9i), her Lp(a) levels decreased from 71.5 to 47.4 mg/dL, and she did not experience further ISR recurrence [69]. While therapeutic strategies targeting Lp(a) are still under development, managing other modifiable cardiovascular risk factors in secondary prevention settings could benefit this population. Recent evidence also indicates that aggressive LDL control may attenuate the adverse cardiovascular effects of elevated Lp(a) [70].

### 3.5. Lp(a) and Cardiac Allograft Vasculopathy

CAV is a progressive narrowing of the allograft vasculature secondary to immunologic and non-immunologic causes [52,71]. The high incidence and morbidity due to CAV greatly hinders the long-term success of heart transplant (HTx) [34] and remains the leading cause of death beyond the first year post-transplantation [35]. Given the impact of CAV on long-term transplant outcomes, it is crucial to identify possible factors contributing to its development. 

The association between Lp(a) and CAV has garnered significant attention recently but still needs to be better understood. Future studies are needed to better define the role of serum Lp(a) in CAV post-transplant. Some potential strengths that can be implemented in future studies to improve accuracy include ensuring the exclusion of donor CAD by performing a baseline angiography within the first months post-HTx and adjusting for medications and other potential confounders, which may play a role in the development of CAV. Given the emergence of Lp(a)-lowering therapies and the formidable barrier CAV poses to transplant success, these studies may play a critical role in identifying patients who may benefit from aggressive Lp(a) control. 

### 3.6. Lp(a) and Bioprosthetic Aortic Valve Degeneration

Calcific aortic stenosis (AS) is the leading valvular heart disease in developed nations, with the prevalence of severe AS in elderly patients estimated to be 3.4% [72]. As life expectancy increases, the number of individuals requiring aortic valve replacement (AVR) is expected to double by 2050, imposing a significant burden on the healthcare systems [72]. Historically, the development of calcific AS was largely attributed to age-related degeneration. However, recent studies have identified that elevated Lp(a) levels were independently associated with accelerated AS progression and an increased risk of AVR [73]. Similar to atherosclerosis, this is thought to involve plasma lipoprotein infiltration triggering active inflammation, the remodeling of the extracellular matrix, and fibroblast proliferation, culminating in pronounced calcification and disease progression [74]. 

AVR is the primary treatment for patients with severe AS, involving the implantation of either a mechanical heart valve or a bioprosthetic valve through surgical or transcatheter approaches. BHV are generally preferred due to their favorable hemodynamic profile and reduced thrombogenicity, and the elimination of the need for lifelong anticoagulation [75]. However, BHVs are susceptible to structural valve degeneration (SVD), an irreversible process that results in intrinsic deterioration of the implanted valve. This is the leading cause of valve replacement failure and frequently necessitates repeat valve surgery. The pathophysiology of SVD is likely multifactorial, involving increased mechanical wall stress, lipid-mediated inflammation, and an immune response to the implanted tissue [76]. The rapidly rising prevalence of BHV implantation heightens the need to ascertain risk factors and associations for SVD, which could inform the development of treatment strategies to minimize this risk. While Lp(a) has been clearly associated with AS progression, its role in bioprosthetic SVD remains unclear. 

The current literature on the association between Lp(a) and bioprosthetic SVD is limited and presents conflicting outcomes, questioning the robustness of this association. Given the variations in the study design and the retrospective nature of both studies, it is difficult to draw definitive conclusions on the topic. The rapid increase in BHV implantation emphasizes the need to gain a clearer understanding of the factors associated with bioprosthetic valve degeneration, as this remains a major challenge in the field. 

Future studies should aim to address these gaps, by performing large, multi-center prospective studies with standardized imaging protocols. Long term follow-up with the incorporation of advanced imaging, such as CT and PET, may help identify early signs of valve degeneration. Clinical trials that explore Lp(a)-lowering medications may help clarify whether Lp(a) levels reduce risk of bioprosthetic SVD. Given the lack of interventions available to prevent or slow the progression of bioprosthetic SVD, clarifying this association may offer a novel potential pathway for reducing the incidence of this troublesome complication [37]. 

## 4. Methods

To explore the association between Lp(a) and the cardiovascular diseases of interest, data from clinical trials, retrospective and prospective observational studies, meta-analyses, and narrative reviews were retrieved MEDLINE/PubMed (U.S. National Library of Medicine, Bethesda, MD, USA), Google Scholar (Google LLC, Mountain View, CA, USA), and Embase (Elsevier, Amsterdam, Netherlands). The search utilized the key words “Lp(a)”; “Lipoprotein (a)”; “Atrial Fibrillation”; “In-Stent Restenosis”; “Cardiac Allograft Vasculopathy”; and “Bioprosthetic Aortic valve degradation”, which were used in different combinations for the years 1990–2024. Additional articles were found by reviewing the references cited in selected publications. Pertinent studies were critically evaluated by the authors, and study inclusion decisions were made by consensus. Non-English publications and those deemed to have low credibility were excluded.

## 5. Conclusions

Elevated Lp(a) levels is a well-recognized risk factor for the development of multiple cardiovascular diseases, including coronary artery disease, aortic stenosis, and cerebrovascular accidents. Novel therapies designed to reduce Lp(a) levels are currently in development and present a promising target in reducing cardiovascular risk. The role of Lp(a) as a risk factor for atrial fibrillation, in-stent restenosis, cardiac allograft vasculopathy, and bioprosthetic valve degeneration remains unclear. While certain studies suggest a potential association, the evidence overall remains inconclusive, with substantial variation in study design, populations, and observed outcomes. The validation of these associations will depend on well-designed, large-scale trials or on their inclusion in prospective studies exploring the benefits of novel therapies targeting Lp(a) reduction.

## Figures and Tables

**Figure 1 ijms-25-11029-f001:**
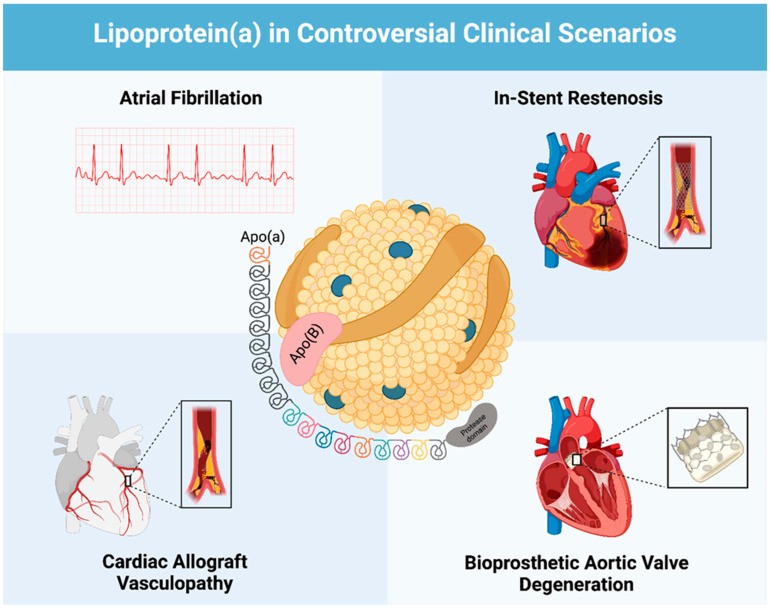
The association of Lipoprotein (a) with atrial fibrillation, in-stent restenosis, cardiac allograft vasculopathy, and bioprosthetic aortic valve degeneration.

**Figure 2 ijms-25-11029-f002:**
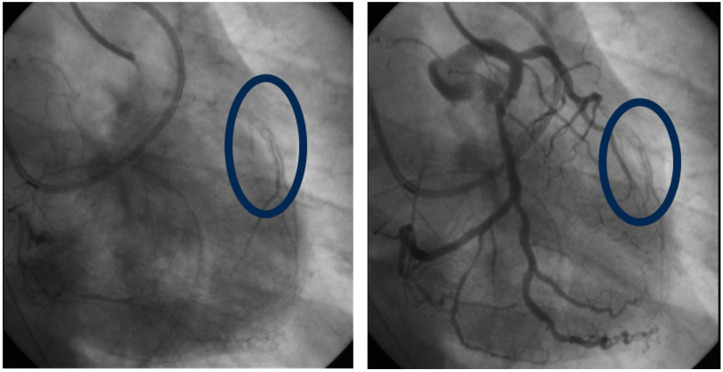
Coronary catheterization images demonstrating left anterior descending artery in-stent restenosis identified in a patient with elevated Lp(a), four years post-stent placement. The circled area highlights the region of restenosis, evidenced by reduced contrast flow through the stented segment.

## Data Availability

Data is contained within the article.

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
