# Peer review of "Lipoprotein (a) as a Cardiovascular Risk Factor in Controversial Clinical Scenarios: A Narrative Review"

_ijms, 2024, doi:10.3390/ijms252011029_

Round 1
Reviewer 1 Report
Comments and Suggestions for Authors
Quantifying the role of lipoproteinemia as a risk factor for various cardiovascular pathologies is necessary and well received, the author's conclusions being pertinent.
There are some minor problems that need to be corrected:
1. Right from the title, the authors must specify the type of review they proposed: systematic or narrative review.
2. In order to fulfill the rigors of a valuable review, it is necessary for the authors to specify what the objectives of the research are.
3. In this sense, it is absolutely necessary to describe the research methodology: the medical databases that were accessed, the studied period, the principle and secondary search lines. The results thus obtained should be included in a table that includes the year of publication of the study, the authors and the main objective.
4. The authors make an excellent description of the relationship between different pathological situations (atrial fibrillation, in-stent restenosis, bioprosthetic aortic valve degeneration, cardiac allograft vasculopathy) and lipoproteins. This particularly valuable information could form the Discussions chapter.
We suggest the authors to make these small structural changes, in order to increase the visibility and impact of the study in the medical community.
Author Response
1. Right from the title, the authors must specify the type of review they proposed: systematic or narrative review.
Completely agree, we have authored a narrative review, ensuring this is reflected in the title.
2. In order to fulfill the rigors of a valuable review, it is necessary for the authors to specify what the objectives of the research are.
3. In this sense, it is absolutely necessary to describe the research methodology: the medical databases that were accessed, the studied period, the principle and secondary search lines. The results thus obtained should be included in a table that includes the year of publication of the study, the authors and the main objective.
Thank you for the valuable suggestion. This will certainly enhance the structure of our manuscript. We have added a methodology section outlining the above points.
4. The authors make an excellent description of the relationship between different pathological situations (atrial fibrillation, in-stent restenosis, bioprosthetic aortic valve degeneration, cardiac allograft vasculopathy) and lipoproteins. This particularly valuable information could form the Discussions chapter.
Great idea. These details have been included in the discussion section, accompanied by sections on Lp(a) structure and function, as well as future directions regarding Lp(a)-lowering therapies.

Reviewer 2 Report
Comments and Suggestions for Authors
Lp(a) lowering strategies seem important for modifying cardiovascular risk. The topic of the review is hot.
Please compare the cardiovascular risk related to cholesterol and Lp(a) in the introduction at least for CAD and stroke.
Is there any effect of antiplatelet therapy on the risk generated by high Lp(a)?
The association between Lp(a) and AF is indirect (though CAD), which probably explains the conflicting results.
CAV has a different pathogenesis. Please consider deleting this pathology in the review, especially since it is not conclusive and the study cohorts are small.
Please add at least in summary whether new RNA therapies would replace statins and modern cholesterol-lowering therapies.
minor
line 103 'tissue regeneration' is a big word, larger than healing. 'Healing' is enough.
Author Response
Please compare the cardiovascular risk related to cholesterol and Lp(a) in the introduction at least for CAD and stroke.
- An important point that warrants discussion, we have incorporated a section on this in the introduction
Is there any effect of antiplatelet therapy on the risk generated by high Lp(a)?
- Another insightful point, and we have included a dedicated section on it in the Lp(a) structure and function section.
The association between Lp(a) and AF is indirect (though CAD), which probably explains the conflicting results.
Yes we agree, we highlighted this discrepancy in the studies in the discussed in section, I have added an additional sentence to further stress this point
- Indeed, we agree and have pointed out this discrepancy in the studies in the discussion section, including an additional sentence to further highlight this
CAV has a different pathogenesis. Please consider deleting this pathology in the review, especially since it is not conclusive and the study cohorts are small.
We recognize that the data and study cohorts are small in this section, but felt it was crucial to discuss this pathology, addressing a large gap in evidence with the goal of prompting further research. We believe this may offer value to readers and hope to retain it, but please inform us if you strongly recommend its removal
Please add at least in summary whether new RNA therapies would replace statins and modern cholesterol-lowering therapies.
- An excellent suggestion that will certainly add value to the article. Please review the section on Lp(a)-lowering therapies in the discussion

Round 2
Reviewer 2 Report
Comments and Suggestions for Authors
The paper was improved. Figures 2 and 4 should be deleted. They don't add anything to the topic.
No further comments.
Author Response
Thank you for your insightful comments and feedback. The figures have been removed accordingly.